# Impact of ITCZ width on global climate: ITCZ-MIP

Angeline G. Pendergrass[1,2], Michael P. Byrne[3], Oliver Watt-Meyer[4,5], Penelope Maher[6], and Mark J. Webb[7]

[1]Cornell University, Ithaca, NY, USA
[2]National Center for Atmospheric Research, Boulder, Colorado, USA
[3]School of Earth and Environmental Sciences, University of St. Andrews, St. Andrews, UK and Department of Physics, University of Oxford, Oxford, UK
[4]Department of Atmospheric Sciences, University of Washington, Seattle, Washington, USA
[5]Allen Institute for Artificial Intelligence, Seattle, Washington, USA
[6]Department of Mathematics, University of Exeter, Exeter, UK
[7]Met Office Hadley Centre, Exeter, UK

**Correspondence:** Angeline G. Pendergrass (a.pendergrass@cornell.edu)

**Abstract.** The width of the Inter-Tropical Convergence Zone (ITCZ) affects tropical rainfall, Earth's albedo, large-scale circulation and climate sensitivity. To better understand the ITCZ width and its effects on global climate, we present a protocol to force changes in ITCZ width in climate models. Starting from an aquaplanet configuration with a slab ocean, adding surface heat fluxes in the deep tropics forces the ITCZ to narrow, and subtracting them causes it to widen. The protocol successfully generates changes in ITCZ width in all four climate models used in this study. ITCZ width in each model responds linearly to forcing magnitude and sign. Comparing across the four climate models, a response to varying ITCZ width that is remarkably consistent among models is the ITCZ strength, which is greater the narrower the ITCZ. On the other hand, the effect of varying ITCZ width on climate sensitivity is divergent among our four models, varying even in sign. Results from this pilot study highlight the connections between surface fluxes, ITCZ width, and the wider climate. A comprehensive Model Intercomparison Project (MIP) has the potential to advance understanding of both the physical processes shaping ITCZ width and its influence on remote atmospheric circulations and global climate.

## 1  Introduction

The Inter-Tropical Convergence Zone (ITCZ) has some of the strongest and most consistent convection and precipitation on the planet. It plays an important role driving the energy and mass fluxes of global atmospheric circulation and is strongly coupled to the Hadley cell – by some definitions, coinciding with its ascending branch. The ITCZ dominates regional hydroclimate in tropical regions, which are vulnerable to even modest changes in its width, strength, or position. Deep convection in the core of the ITCZ drives teleconnections via Rossby wave trains that propagate to the midlatitudes of the Northern and Southern Hemispheres, thus potentially affecting many aspects of global climate, including the width of the tropics and the location of the storm tracks (Watt-Meyer and Frierson, 2019).

The width of the ITCZ and its changes in width may also influence the response to global warming beyond the deep tropics, potentially influencing global precipitation (Su et al., 2017) and climate sensitivity (Barsugli et al., 2005; Tian, 2015; Webb and

Lock, 2020). Fundamental understanding of the changes in ITCZ width and its effects on global climate is therefore important. However, until recently there has been relatively little research on it, other than the double ITCZ bias in climate models, which we set aside for this study (reasoning for which is discussed in Section 2.2.1). There are indications that the width of the ITCZ may decrease in response to global warming; in climate model projections forced by increasing greenhouse gases, the ITCZ narrows (Byrne and Schneider, 2016a; Byrne et al., 2018), and observational analysis has revealed evidence of decreasing ITCZ width in some ocean basins (Wodzicki and Rapp, 2016). Existing literature on what controls the width of the ITCZ, including energetic and potential dynamical theories, was synthesized by Byrne et al. (2018). Other explanations for the ITCZ width and its variations have since been put forward, including aggregation associated with zonal asymmetries (Popp and Bony, 2019), cloud radiative effects (Dixit et al. 2018; see also Harrop and Hartmann 2016), Ekman balance in the tropical boundary layer (Byrne and Thomas, 2019), and variability in the seasonal cycle of ITCZ location (Donohoe et al., 2019). Most of these studies focused on experiments using a single climate model.

Despite that current comprehensive climate models share many similarities, comparing simulations from different models can be useful, especially when they reveal different yet plausible emergent behaviors. Of particular relevance to the ITCZ, tropical clouds and precipitation have dramatically different responses to increasing greenhouse gases in different climate models, even in simulations with an aquaplanet configuration (Stevens and Bony, 2013), which is an idealized configuration with no continents or seasonal cycle. Developing understanding of ITCZ width that is robust across climate models thus calls for a Model Intercomparison Project (MIP).

In response to this calling, and in order to investigate the effects of ITCZ width on global climate, we introduce ITCZ-MIP. The experimental protocol, described in the next section, is based around simulations run in aquaplanet slab-ocean configurations, in which the ITCZ is forced to change via additional heat fluxes applied at the surface. A pilot set of 4 models, each in aquaplanet configuration, serves to demonstrate and validate the protocol. Three of the models are standard CMIP-class models and one is a more idealized model without cloud radiative effects. Two unanticipated behaviors emerge in the pilot MIP: first, a relationship between ITCZ width and global-mean temperature (in the absence of external forcing; Section 3.2); and second, a remarkably consistent relationship between ITCZ width and strength (Section 3.3). The pilot simulations enable an exploratory analysis of two specific aspects of how ITCZ width affects global climate (Section 4): (1) the relationship between ITCZ width and Hadley cell width and the location of the storm tracks, and (2) the relationship of ITCZ width to equilibrium climate sensitivity. Finally, we reflect on key learnings thus far and new questions prompted by the pilot MIP.

## 2   ITCZ-MIP Experimental Protocol Description

In this section we present the experimental protocol: the approach to generate changes in the width of the ITCZ. Then we document the metrics we use to quantify ITCZ width and describe the model simulations in the pilot MIP.

## 2.1 Aquaplanet configuration with no seasonal cycle

Aquaplanet configurations are at an intermediate level in the model hierarchy. They typically have comprehensive atmospheric models, including full radiative transfer, parameterized clouds and parameterized convection, as well as comprehensive dynamical cores (the same as the parent climate model). What defines an aquaplanet is the reduced complexity boundary conditions. Aquaplanets do not have continents (making these models zonally symmetric). As far as the seasonal cycle of solar insolation and its many effects, it is possible to include the seasonal cycle in aquaplanet simulations (e.g., Voigt et al., 2016), though often it is excluded. For surface boundary conditions, atmosphere-focused aquaplanet configurations typically have an idealized representation of the ocean, using prescribed SSTs or a slab ocean. The slab ocean indirectly represents ocean dynamics by prescribing vertical heat fluxes in each ocean surface grid box, while still representing the thermal inertia of the ocean and exchanging surface fluxes of water and heat. Surface heat flux can be added to slab ocean as an idealized representation of horizontal heat transport within the ocean. Aquaplanet models are an important model configuration within climate model hierarchies and are an extremely useful model configuration for asking fundamental science questions (Maher et al., 2019), such as we are asking in ITCZ-MIP.

Interactions between the ocean and atmosphere are fundamental to the climate, particularly in the tropics. But fully coupled ocean-atmosphere simulations are expensive and a step towards the comprehensive end of the model hierarchy. We choose to focus ITCZ-MIP on a middle ground: the slab ocean (also known as mixed-layer ocean, e.g., Stouffer and Manabe, 1999), in which each grid cell has prognostic sea surface temperature, a fixed heat capacity, and prescribed $q$-fluxes representing convergence of ocean heat transport that would arise from ocean currents.

Despite their simplifications, when different climate models are run in aquaplanet configuration, they can still have dramatically different behavior - especially in terms of clouds, precipitation, and their responses to greenhouse gas forcing (Stevens and Bony, 2013). The differences among aquaplanet simulations often contain the relevant mechanisms to their parent Earth-configuration models in terms of their climate and sensitivity to warming (Medeiros et al., 2008, 2015; Ringer et al., 2014).

The ITCZ-MIP experimental protocol focuses around a series of aquaplanet simulations coupled to a slab ocean. Simulations follow the Aqua-Planet Experiment (APE)-protocol of Neale and Hoskins (2000) and Williamson et al. (2012). Each simulation has perpetual equinoctial solar insolation, solar constant equal to 1365 W m$^{-2}$, a diurnal cycle, $CO_2$ concentrations of 348 (or 1392 ppm for quadrupled $CO_2$), zonally symmetric latitude-height ozone (as in Williamson et al., 2012). All simulations whose output is intended to be analyzed have a slab ocean with a 10-m mixed layer depth; one simulation with prescribed SST is needed as part of the standard setup of slab ocean simulations (described in the next section). Ten meters is a mid-range depth for a slab ocean; small differences in choice of depth would not obviously behave qualitatively differently (e.g., Donohoe et al., 2014), though larger slab-ocean depths require longer spin up time. A full radiation scheme is recommended for ITCZ-MIP in which the $CO_2$ concentration can be directly specified; each model in this study has a full radiation scheme. An alternative option is to use a gray radiation scheme and implement an equivalent $CO_2$ concentration as in, e.g., TRAC-MIP (Voigt et al., 2016).

Recent work showing that the seasonal cycle may be important for observed annual-mean ITCZ width (Donohoe et al., 2019; Zhou et al., 2020). Meanwhile, in observations of the zonal mean distribution of daily precipitation amount, the seasonal cycle of the tropical rain belts over ocean is largely a modulation of precipitation amount (and frequency) rather than its intensity of location; this contrasts markedly with the situation of tropical land, where the seasonal cycle is dominated by migration with the sun (Fig. 5, Pendergrass and Deser, 2017). Furthermore, it has been shown that even configurations with even more

idealizations beyond removing the seasonal cycle are relevant to the tropical atmosphere (Popke et al., 2013). Including a seasonal cycle adds physical complexity to the ITCZ width, in particular the potential for interactions between ITCZ width and location. While these interactions may be relevant for annual-mean ITCZ width (Zhou et al., 2019), isolating the effects of ITCZ width from location provides an opportunity to break a complex behavior into its component parts, which we can hope might be simpler and feasible to understand in more depth. For these reasons, we choose to follow the common practice of

setting the solar insolation to equinox conditions, with the sun directly overhead at the equator. Rather than representing annual mean conditions, this is best thought of as a perpetual single season. We expect that future work would be needed to integrate learnings from this MIP back up the model hierarchy towards the full complexity of the real atmosphere on Earth.

## 2.2 Control experiments

In order to generate the $q$-fluxes to run the slab ocean simulations, a control simulation is first run using prescribed SSTs.

The resulting $q$-fluxes are then used as input for two slab-ocean control simulations, one for each $CO_2$ concentration. This subsection discusses the choice of SST profile to prescribe, followed by describing the procedure to calculate the $q$-fluxes.

### 2.2.1 A single ITCZ

As observed in nature, there is one primary band of maximum precipitation in the annual mean at most longitudes in the tropical ocean basins - a single ITCZ - which is on average offset toward the northern hemisphere. A persistent bias common to most

climate models, referred to as the double ITCZ bias, is for models to have two precipitation maxima of similar magnitude, one in each hemisphere. The double ITCZ bias has persisted for decades over generations of climate models. Recent mechanisms put forward that may be responsible for this bias are atmosphere-ocean feedbacks and biased strength of regional patterns of seasonal convection (e.g., Song and Zhang, 2019; Song et al., 2019).

The double-ITCZ bias in climate models is an interesting problem in its own right, and it might be in some sense related to the

width of the ITCZ in full present-day geography settings. However, we believe there is much still to learn about the processes shaping the width of a single ITCZ and its influence on climate. Therefore, we choose to focus on idealized experimental configurations that result, for the most part, in just one zonal-mean precipitation maximum in the tropics. We accomplish this by starting from an initial prescribed SST profile that generates a single ITCZ.

The prescribed-SST control simulation (itcz-SST in Table 1) is generated with the APE-protocol's Control prescribed-SST profile (Neale and Hoskins, 2000). The Control SST profile is described by Eq. 1,

$$\text{SST}(\phi) = \begin{cases} 27\left(1 - \sin^2\left(\frac{3\phi}{2}\right)\right){}^\circ C, & \text{if } -\frac{\pi}{3} < \phi < \frac{\pi}{3}, \\ 0{}^\circ C, & \text{otherwise.} \end{cases} \tag{1}$$

The Control SST profile differs from the default "Qobs" profile used in e.g., the CFMIP CMIP6 simulations (Webb et al., 2017) in that the Control SST profile has a narrower peak. This narrower peak in SST also results in narrower ITCZ (Rajendran et al., 2013, not shown here), and avoids a split ITCZ in the control state.

### 2.2.2 Slab ocean and prescribed $q$-fluxes

The process for running the slab ocean simulations in this protocol builds on the prescribed-SST control simulations. Output from the itcz-SST control simulation is used to calculate the climatological surface energy imbalance ($F$), from which $q$-fluxes are generated to run all slab-ocean experiments, i.e., Experiments 1-8 in Table 1. The surface imbalance is the sum of radiative and turbulent heat fluxes,

$$F = SW + LW - LH - SH - SN, \tag{2}$$

where $SW = SW_\downarrow - SW_\uparrow$ is the net shortwave radiative flux at the surface, $LW = LW_\downarrow - LW_\uparrow$ is the net surface longwave radiative flux, $LH$ is the latent heat flux associated with evaporation, $SH$ the sensible heat flux and $SN$ the latent heat flux associated with frozen precipitation. The up and down arrows correspond to upward and downward fluxes, respectively. Positive values of $F$ indicate excess downward energy at the surface (i.e., the surface temperatures would be warmer than the specified SST if they were allowed to vary). If $LH$ and $SN$ are not directly output by a climate model, they can instead be calculated using the surface evaporation and frozen precipitation using the appropriate latent heat of evaporation and latent heat of fusion. The global mean $F$ does not explicitly need to equal zero, but $F$ should be very similar to the global mean net TOA radiative flux in order for the atmosphere to stay close to energetic balance (this is also true for the specified-SST simulation). The present-day $CO_2$ slab-ocean control simulation itcz-slab (Experiment 1) and increased $CO_2$ control itcz-slab-4xCO2 (Experiment 6) need only this base-state $q$-flux forcing $F$ to run.

### 2.3 Additional $q$-flux forcing to change ITCZ width

Experiments where ITCZ width varies within one climate model at a time distill the essence of the ITCZ width variation and its effects on global climate. Next, we come to the heart of the ITCZ-MIP protocol: applying additional $q$-fluxes to generate variations in ITCZ width. The additional $q$-flux ($q_{itcz}$) has the shape of the top half of a sinusoid in latitude, reaching a maximum at the equator, with compensating energy flux of the opposite sign poleward of the subtropics to balance the added

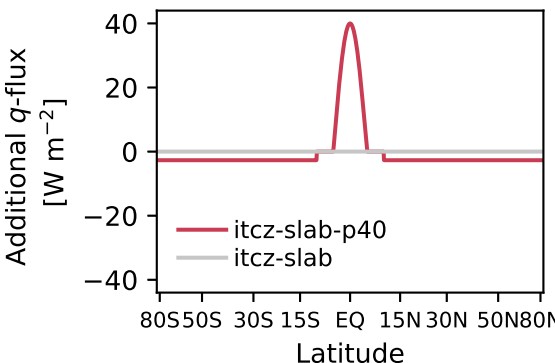

**Figure 1.** The additional $q$-flux forcing with $A = 40$ W m$^{-2}$, described by Eq. (3).

heat flux in the global mean.

$$
q_{itcz}(\phi) = \begin{cases} A\cos\left(\frac{\pi}{2}\frac{\phi}{\Delta\phi}\right), & |\phi| < \Delta\phi \\ 0 & \Delta\phi < |\phi| < 2\Delta\phi \\ B, & |\phi| > 2\Delta\phi, \end{cases} \tag{3}
$$

where $A$ is the magnitude of heating at the equatorial peak, and $\Delta\phi$ is the width of the anomalous heating at the equator ($\Delta\phi = 5°$). The compensation applied evenly poleward of the subtropics determined by the term $B$ is given by Eq. (4) and

is computed with the condition that the global-mean $q_{itcz}$ is zero. This is achieved by setting $\int_{-\pi/2}^{\pi/2} q_{itcz}(\phi)\cos(\phi)\,d\phi = 0$. Specifically,

$$
B = \frac{A\cos(\Delta\phi)}{2\left(1 - \sin(2\Delta\phi)\right)\left(\frac{\Delta\phi}{\pi} - \frac{\pi}{4\Delta\phi}\right)}. \tag{4}
$$

The total $q$-flux ($q$) for each experiment is,

$$
q = -F + q_{itcz}, \tag{5}
$$

where $F$ is given by Eq. (2) and $q_{itcz}$ is given by Eq. (3). We note that when $A = 0$ in Eq. (3), the total $q$-flux is given by $q = -F$. The $q_{itcz}$ with $A = 40$ W m$^{-2}$ for the itcz-slab-p40 experiment is shown in Fig. 1.

### 2.4 Experiment collection with varying ITCZ width

In order to test the linearity of variations in ITCZ width and climate responses to forcing, two positive and negative $A$ values are applied in separate experiments: $A = -40, -20, 20, 40$ W m$^{-2}$. An analogous set of quadrupled carbon dioxide simulations

enables examination of the effect of base-climate ITCZ width on global warming.

**Table 1.** ITCZ-MIP experiments. Tier I experiments are essential and Tier II experiments are highly encouraged. The scaling factor $A$ is used in Eq. (3) to set the ITCZ width.

| No. | Experiment name | Description | $A$ [W m$^{-2}$] | $CO_2$ [ppm] |
|---|---|---|---|---|
| Tier I experiments | | | | |
| 0 | itcz-SST | SST profile using Eq. (1) | N/A | 348 |
| 1 | itcz-slab | Slab ocean control | 0 | 348 |
| 2 | itcz-slab-m20 | As in 1 and a wider ITCZ | -20 | 348 |
| 3 | itcz-slab-p20 | As in 1 and a narrower ITCZ | +20 | 348 |
| Tier II experiments | | | | |
| 4 | itcz-slab-m40 | As in 2 and a wider ITCZ | -40 | 348 |
| 5 | itcz-slab-p40 | As in 3 and a narrower ITCZ | +40 | 348 |
| 6 | itcz-slab-4xCO2 | As in 1 and $4\times CO_2$ | 0 | 1392 |
| 8 | itcz-slab-4xCO2-m20 | As in 2 and $4\times CO_2$ | -20 | 1392 |
| 9 | itcz-slab-4xCO2-p20 | As in 3 and $4\times CO_2$ | +20 | 1392 |
| 7 | itcz-slab-4xCO2-m40 | As in 4 and $4\times CO_2$ | -40 | 1392 |
| 8 | itcz-slab-4xCO2-p40 | As in 5 and $4\times CO_2$ | +40 | 1392 |

The experiments that constitute ITCZ-MIP are listed in Table 1. They are broken down into two tiers, analogous to the CMIP protocol. The Tier I experiments are requested from all contributing models, and Tier II are highly encouraged (but not essential for contributing models to ITCZ-MIP). Each experiment includes 30 years of simulations after removing a suitable period to allow for spin up (the spin up will vary between models, and may be up to five years).

The variables requested as part of ITCZ-MIP are listed in Table 2. The procedure for ITCZ-MIP is summarized as follows:

1. Generate the control experiment (itcz-SST) using the SST profile in Eq. (1).

2. Compute the $q$-flux ($q_{itcz}$) and climatological zonal-mean surface energy imbalance ($F$) from the itcz-SST run using Eq. (2–Eq. 5) for the neutral ITCZ width ($A = 0$).

3. Impose the total $q$-flux ($q$) from Eq. (5) in the slab ocean experiment to generate itcz-slab. Other than these changes to a slab-ocean boundary condition, the same parameters and settings should be used as in itcz-SST.

4. Repeat step 3 for two positive deep tropical $q$-flux forcing corresponding to a narrower ITCZ ($A = 20, 40$ W m$^{-2}$) and two negative forcing (wider ITCZ, $A = -20, -40$ W m$^{-2}$) setups.

5. Repeat step 4 using quadrupled $CO_2$ concentration (1392 ppm).

**Table 2.** Model output variables and frequency requested for all of the experiments listed in Table 1. The variable names listed are standard CMIP variable names. All output should follow CMIP6 standards, i.e. in terms of dimensional names and units.

|  | Variable name | Description |
|---|---|---|
| Tier I; Monthly 3D | ua, va, wap, ta, hur, hus, zg | winds, air temperature, relative humidity, specific humidity and geopotential height |
| Tier I; Monthly 2D | uas, vas, tauu, tauv | surface winds and wind stresses |
|  | tas, huss | near-surface temperature and specific humidity |
|  | pr, prc, prsn | precipitation, convective precipitation, snow |
|  | ts, ps | SST, surface pressure |
|  | prw, clt, evspsbl | liquid water path, cloud fraction, evaporation |
|  | hfls, hfss | Sensible and latent heat flux |
|  | rlds, rlus, rsds, rsus, rsdt, rsut, rlut | All-sky radiative fluxes |
|  | rldscs, rluscs, rsdscs, rsuscs, rsutcs, rlutcs | Clear-sky radiative fluxes |
|  | $q$-flux | The imposed $q$-flux from Eq. (5) |
| Tier I; Daily 2D | pr, prc, evspsbl, clt, tas, huss, wap500 | As above but for daily data |
| Tier II; Monthly 3D | usvs, vsts, vsqs | Eddy fluxes of momentum, heat, moisture |
| Tier II;Daily 3D | ua, va, ta, hus, zg, wap | As above but for daily data |

## 2.5 Pilot ITCZ-MIP models

We run the ITCZ-MIP protocol with four models to demonstrate that the experimental approach does indeed result in varying ITCZ width in a handful of different models, and to highlight a few interesting aspects of the climate responses and motivate contributions with more models. The goal of this pilot study is to lay a framework that increases the likelihood of success for a full-scale MIP.

The pilot MIP include three CMIP-class models, and one more idealized model. CESM2.0 simulations (which have the
CAM6 atmosphere component and physics package, Danabasoglu et al., 2020) use the finite volume grid at nominal 1 degree resolution. Two sets of simulations have a nominal 2-degree resolution: CESM1.2 (CAM5 atmosphere, Neale et al., 2012; Meehl et al., 2013) and GFDL-AM2.1 (Delworth et al., 2006). The fourth model in the pilot MIP is Isca, an idealized modelling framework for the global circulation of Earth and other planetary atmospheres (Vallis, G. K. et al., 2018). Isca simulations are run at T42 (2.8°×2.8°) resolution with 41 vertical levels (0.027 hPa top). They use the comprehensive SOCRATES radiation
scheme and all other parameterizations as standard for the GFDL FMS: simple Betts-Miller convection, no clouds, and simple boundary layer scheme.

## 2.6 ITCZ width metric

In order to examine variations in ITCZ width quantitatively, it is necessary to choose a metric. In the aquaplanet configuration, with no zonal-mean asymmetry or seasonal cycle and just one maximum of zonal-mean precipitation in the tropics (by design,
see Section 2.2.1) in most simulations, this is relatively straightforward. Nonetheless, there are multiple ways to quantify ITCZ width, some of which are described in Byrne et al. (2018). These include a mass-based definition, which is the meridional distance between extrema of the streamfunction that bound the region of mean ascending flow, and a moisture-based definition, which is the meridional distance between zero-crossings of tropical precipitation minus evaporation. We focus on the "mass" ITCZ width; the distance between latitudes closest to the equator with a zero-crossing of the meridional derivative of the 500
hPa streamfunction. In this manuscript, we express the distance in degrees latitude.

## 3 Results: Validating the protocol

In this section we demonstrate the successful implementation of the protocol in multiple modeling frameworks, and then explore potential limitations of the experiments.

### 3.1 ITCZ width variations driven by prescribed $q$-fluxes

First, we verify that the prescribed $q$-fluxes result in changes in ITCZ width. Figure 2 shows the total zonal-mean precipitation for each of the $q$-flux forcings, as well as their differences from the itcz-slab control experiment for one model (CESM2). The $q$-flux forcing drives a systematic change in the width of the ITCZ, as intended; heating of the deep tropics leads to narrowing of the ITCZ ("p" for positive deep tropical heating experiments), while cooling of the deep tropics leads to ITCZ widening

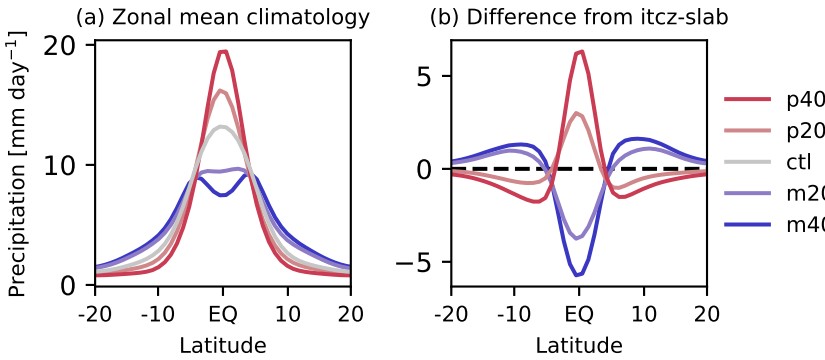

**Figure 2.** Climatological zonal-mean precipitation for (a) Experiments 1-5 and (b) Experiment 2-5 minus itcz-slab (Experiment 1) for CESM2.

("m" experiments). This is visible in terms of the width of the deep precipitation belt (Fig. 2a), and is also reflected in the mass-ITCZ-width metric based on the 500 hPa streamfunction (Fig. 3, see section 2.6, and consistent with other metrics like $P - E$-based width, not shown). Changes in ITCZ width also coincide with changes in peak precipitation strength (Fig. 2); a narrower ITCZ coincides with an increased peak zonal-mean precipitation on and near the equator and reduced precipitation in the rest of the tropics (and the reverse for a widening ITCZ). In the widening ITCZ experiments (especially the one with the largest forcing, m40), the ITCZ splits into separate local zonal-mean precipitation maxima in each hemisphere in some models, including CESM2 (blue line, Fig. 2a).

One aspect that the protocol is designed to address is the linearity of the ITCZ width variations to magnitudes and sign of $q$-flux forcing. To accomplish this, the experimental protocol has two magnitudes of $q$-flux forcing that are relatively large locally near the equator, one a factor of 2 larger than the other (peaking at 20 and 40 $\mathrm{Wm}^{-2}$). These $q$-flux forcings are each applied with positive and negative sign. These constitute four experiments in the current base climate, and a fifth which is a control (no $q$-flux forcing).

The width of the ITCZ as a function of the magnitude of $q$-flux forcing magnitude and sign for the four pilot MIP models is shown in Fig. 3. The ITCZ width response is remarkably linear for all four models, though the magnitude of the variation across the experiments varies among the models (the slopes of the lines in Fig. 3; listed in Table 3).

### 3.2 Global mean temperature and ITCZ width

The $q$-flux forcing applied in these experiments adds heat to the system in the deep tropics. In order to avoid directly adding net energy in the global mean, the same amount of energy is removed at higher latitudes in each hemisphere, such that the net global addition of heat is exactly zero. Therefore, if the only factor determining global mean temperature were the applied forcing, then we would expect global-mean temperature to remain unchanged across the experiments.

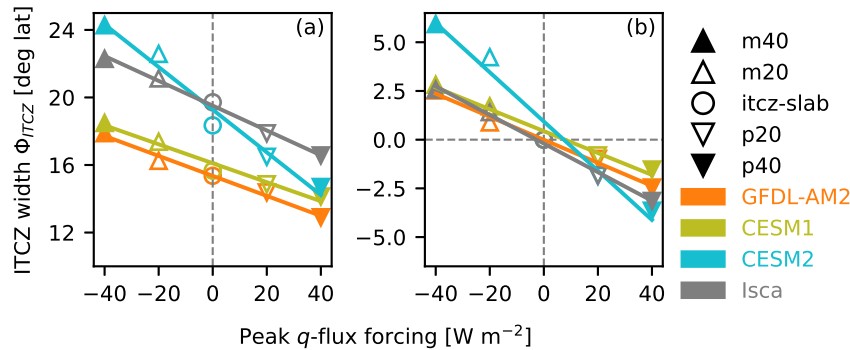

**Figure 3.** ITCZ mass width as a function of peak $q$-flux forcing in the control simulations, with present-day $CO_2$, for all models. (a) Absolute width and (b) width difference from itcz-slab (Experiment 1). ITCZ mass width metric is based on the zero crossing of the 500 hPa streamfunction (section 2.6). A regression line is included for each model

**Table 3.** Slope of regression line to ITCZ width versus $q$-flux forcing magnitude for current $CO_2$ levels in each model.

| Model | Slope |
|---|---|
| GFDL-AM2.1 | -0.059 $^\circ lat/$W m$^{-2}$ |
| CESM1.2 | -0.056 $^\circ lat/$W m$^{-2}$ |
| CESM2.0 | -0.126 $^\circ lat/$W m$^{-2}$ |
| Isca | -0.073 $^\circ lat/$W m$^{-2}$ |

This is, however, not quite the case. Figure 4a shows the global-mean temperature versus ITCZ width for each model and $q$-flux experiment (all with baseline $CO_2$ concentrations, and using the mass-based width metric). There is a nearly linear relationship between ITCZ width and temperature across experiments for each model; for all models, temperature increases with widening ITCZ across the experiments. To a large extent, the magnitude of the temperature versus control-ITCZ width-slope is also similar among the three comprehensive models; recall that Isca has no representation of cloud radiative effects. In order to confidently move forward with using this MIP protocol to probing questions about how ITCZ width affects global climate, we first attempt to understand why global mean temperature responds to our protocol despite not adding any energy to the system in the global mean.

There are at least two possible mechanisms that could lead to the relationship between global mean temperature and ITCZ width, each of which we consider in turn. The first is the "radiator fin" mechanism from Pierrehumbert (1995). This mechanism hinges on the notion that cooling to space varies with global temperature most strongly in the subtropical descending branch of the Hadley circulation. In the ascending branch, there is strong compensation between LW and SW cloud feedbacks, and so sensitivity to climate state is small. Pierrehumbert (1995) developed an heuristic model in which global surface temperature

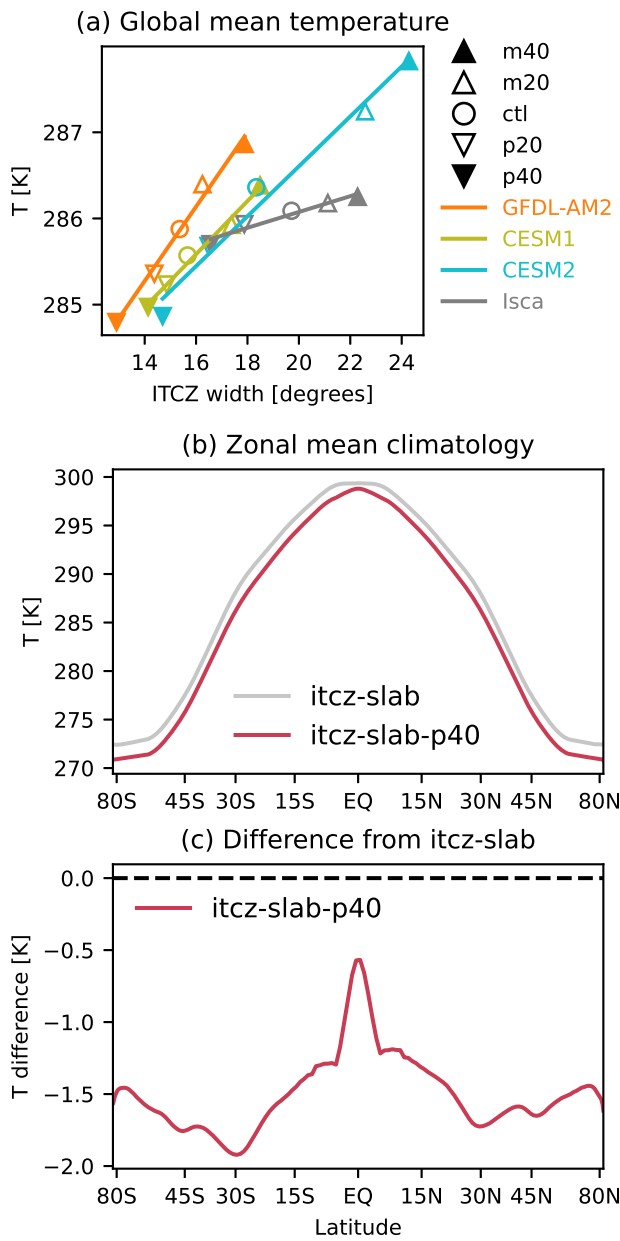

**Figure 4.** (a) Global mean surface temperature versus ITCZ width in ITCZ MIP experiments, including a regression line for each model. Zonal mean temperature in CESM2 control and decreased (narrow) ITCZ width experiments (itcz-slab-p40); (b) zonal-mean total temperature and (c) difference from control.

decreases as the fractional area of descent increases, because this increases the cooling efficiency of the planet. This would be consistent with higher global-mean temperature for a wider ITCZ.

The ITCZ-MIP pilot models' behavior (Fig. 4a) is consistent with the prediction of Pierrehumbert's radiator fin model. Furthermore, the radiator-fin effect is apparent in the OLR field of ITCZ-MIP experiments (Fig. 5). In the control experiment (Fig. 5a; dashed blue line), OLR peaks just poleward of the latitude where the descending region begins. As the ITCZ narrows (in the p40 experiment; solid blue line), the descending branch extends (or perhaps shifts) equatorward, and the peak OLR increases in strength. The resulting change in TOA net LW (Fig. 5b, solid blue line; note the sign has changed to downwelling positive in preparation for further comparisons below), is a lobe of decreasing downwelling LW (increasing OLR) in the subtropics, which corresponds to a loss of energy by the atmosphere. Coincident with the radiator-fin OLR response, there is also a change in net downwelling SW flux at the TOA which partially compensates; but overall the total change in flux is dominated by the OLR, radiator-fin response at the edge of the descending branch abutting the ITCZ. Within the ITCZ itself, the region in Fig. 5 where $\omega_{500}$ is upward, larger magnitude changes in both SW and LW flux partially compensate, which is an essential component of the radiator fin mechanism. This compensation occurs to varying degrees among the ITCZ-MIP models. Overall, as the ITCZ narrows, the tropics lose more energy to space via radiative flux, while the Earth gains energy in the extratropics. This effect can be viewed as one aspect of the response to changing ITCZ width, since the equatorward expansion of the descent region is what fills the area that has ascent in the control simulation.

A second potential mechanism offered by the literature is the variation in efficacy of ocean heat uptake on surface temperature with latitude (Kang and Xie, 2014; Rose et al., 2014). These studies showed that when the same $q$-flux (interpreted as ocean heat uptake in these studies, but nonetheless analogous) is applied at high and low latitudes, global mean temperature increases far more in response to the high-latitude forcing than the low-latitude forcing. As our experimental design is to have a $q$-flux that integrates globally to zero by compensating deep tropical heating with cooling outside of the deep tropics, the difference in efficacy of these two types of forcing probably contributes to the overall increase in global mean surface temperature. In Fig. 4c, we see that for this experiment where heat is added at the equator and removed in mid- and high-latitudes, near-surface air temperature cools, consistent with the differing efficacy mechanism. In CESM2 (Fig. 4c), the peak cooling occurs near 30° latitude, and polar amplification is modest.

Interestingly, while Kang and Xie (2014) did not specifically analyze the width of the ITCZ in their simulations, they did report on the strength of the Hadley circulation, which (as we will see in the next section) is closely linked to ITCZ width. In particular, heating in the tropics strengthens the Hadley cell and corresponds to a narrowing of the ITCZ, while heating the extratropics weakens the Hadley cell, probably corresponding to a widening of the ITCZ. The ITCZ-MIP prescribed forcing can be thought of as a superposition of tropical heating and extratropical cooling (or vice versa), and to the extent that the Hadley cell responses are additive and correspond to ITCZ width changes, the variations in ITCZ width we would infer from Kang and Xie (2014) agree in sign with our experiments. Furthermore, the change in net TOA fluxes in the tropics and extratropics in our experiments are also consistent with their framework.

Other studies have also noted nonlinear responses to regional warming and cooling perturbations. For example, Shaw et al. (2015) applied warming and cooling perturbations with zero zonal mean to slab aquaplanet experiments with the MPI-ESM-LR model. They found that this shifted the ITCZ, a behaviour that wasn't predicted by previous energetic arguments. Cloud locking experiments attributed this behaviour to a nonlinear response of the shortwave cloud radiative effect to local warming

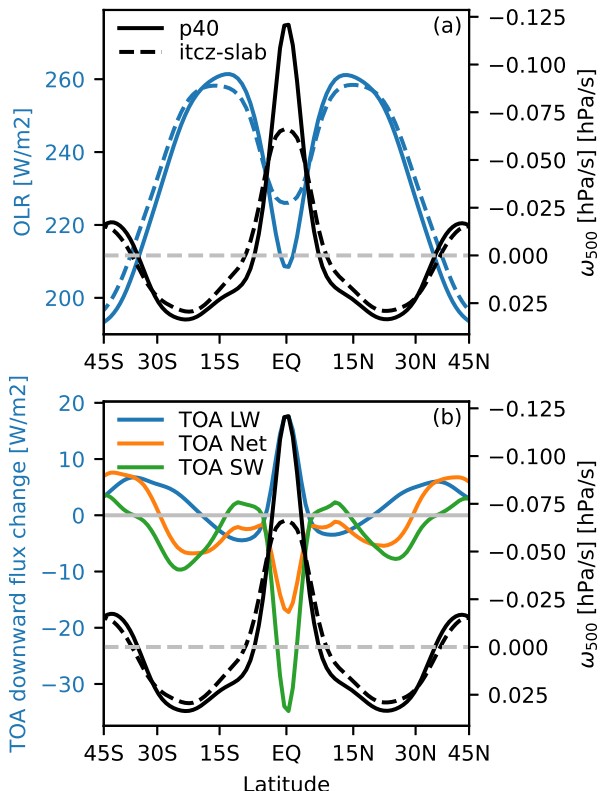

**Figure 5.** TOA radiative fluxes and $\omega_{500}$ for itcz-slab-p40 compared to itcz-slab experiments for CESM2. (a) OLR and $\omega_{500}$ in each simulation, focusing on the tropics. (b) Changes in downwelling LW, SW, and net radiative fluxes at the TOA and $\omega_{500}$ for itcz-slab and itcz-slab-p40 simulations.

and cooling. Shin et al. (2017) found similar nonlinearities in cloud responses in the GFDL AM2 model in a study which
examined the impact of warming and cooling perturbations at different latitudes on the Arctic. Hence a nonlinearity in the
responses to local warming and cooling perturbations is present in GFDL AM2, the model with the strongest response across
forcings in Fig. 4. This adds further weights to the ocean heat uptake efficacy argument, for AM2 at least.

The implications for interpreting ITCZ-MIP experiments are that the compensating $q$-fluxes in the extra-tropics, included to
avoid directly adding energy in the global-mean and thus a net climate forcing, in some sense might produce their own impacts
on global climate, which it may not be appropriate to view as impacts of ITCZ width variations. On the other hand, the changes
in Hadley cell in Kang and Xie (2014) could also be seen as another pathway through which the additional $q$-fluxes actually
force the ITCZ width to change. Therefore, we proceed with appropriate caution, bearing this ambiguity in mind.

In summary, there are at least two mechanisms which could explain the unanticipated relationship between ITCZ width and
global mean surface temperature for our itcz-slab experiments: the increase in LW cooling to space as the area of the descent
region increases (the radiator fin mechanism) and the higher efficacy of extratropical relative to tropical ocean heat uptake.

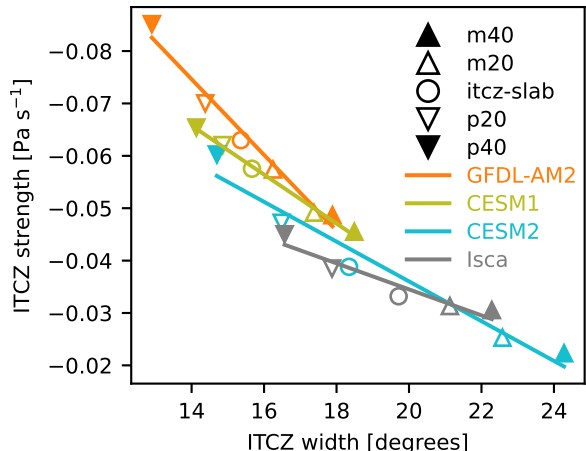

**Figure 6.** Strength versus width of the ITCZ for the present-day $CO_2$ simulations of all models.

### 3.3 ITCZ strength and width

A key role of the ITCZ in global circulation is that it is the location of mean ascent into the tropical upper troposphere. The strength of the ITCZ, in terms of the mass-flow of air, is a key driver of global circulation. In climate model simulations of global warming, the change in strength is anti-correlated to the projected change in ITCZ width across models, though slightly more models project decreasing than increasing ITCZ strength (Byrne et al., 2018). In this section, we document the relationship between strength and width in the pilot ITCZ-MIP simulations.

We quantify the ITCZ strength as the total mass transport divided by total area, or the bulk vertical pressure velocity for the ITCZ $\omega_{ITCZ}$ following Byrne et al. (2018),

$$\omega_{ITCZ} = -g\Psi_{ITCZ}/A_{ITCZ}, \tag{6}$$

where $\Psi_{ITCZ}$ is the mass streamfunction and $A_{ITCZ}$ is the area of the ITCZ. Among the ITCZ-MIP experiments with control $CO_2$ concentration, ITCZ strength and width have a consistent anti-correlation: the narrower the ITCZ, the stronger it is, both across experiments within each model and also among models (Fig. 6).

One might be concerned that this relationship could be more an inadvertent effect of the $q$-flux forcing that drives the variation in ITCZ width in our experiment, rather than a fundamental relationship between ITCZ width and strength themselves, since adding heat directly to the ITCZ could increase ITCZ strength as well as ITCZ width. However, even setting aside the experiments with $q$-fluxes added to change ITCZ width and instead comparing among control simulations in different models, the strong inverse relationship between strength and width remains. This preliminary finding prompts an intriguing question that could be investigated with a full-scale ITCZ-MIP containing more climate models: is the inverse relationship between ITCZ strength and width robust within and across climate models? If so, what is its mechanism?

## 4   Impact of ITCZ width on climate

One motivation for the ITCZ-MIP is to examine the effect that variations of ITCZ width have on other aspects of climate. In this section we document some initial findings in the pilot ITCZ-MIP simulations. First, we consider the response to varying ITCZ width while holding $CO_2$ concentrations fixed; then, we examine the effects on climate as $CO_2$ concentrations increase.

### 4.1   Effects of ITCZ width on Hadley cell width and midlatitude jet position

Tropical deep convection is the engine of the global atmospheric circulation, so we might expect that changes in the width of this convection could affect circulation at higher latitudes. In all four models of the pilot MIP, a narrower ITCZ is associated with an overall narrower Hadley cell (Figs. 7 and 8). Furthermore, it is also associated with an equatorward shift in the eddy-driven jet (Fig. 9). These circulation responses, which coincide with a strengthening of the subtropical jet (Fig. 9) could be explained as follows. A narrower ITCZ implies greater poleward transport of zonal angular momentum in the upper troposphere, and hence a stronger subtropical jet. The stronger subtropical jet in turn leads to increased baroclinicity at lower latitudes and therefore an equatorward shifted Hadley cell edge and eddy-driven jet (Lee and Kim, 2003; Brayshaw et al., 2008; Watt-Meyer and Frierson, 2019). The responses in all four pilot ITCZ-MIP models are both broadly consistent with the findings from experiments with the GFDL AM2.1 model in an aquaplanet configuration with prescribed SSTs, in which varying deep tropical SSTs induced changes in ITCZ width, Hadley cell width, and eddy-driven jet location (Watt-Meyer and Frierson, 2019).

### 4.2   Increased $CO_2$

Another motivation for investigating ITCZ width is the indication of its narrowing in simulations of global warming in comprehensive climate models and in observations of the real world. Webb and Lock (2020) posed the hypothesis that deep convection encroaching into low-cloud regions disrupts low clouds and diminishes positive low-cloud feedbacks, resulting in smaller climate sensitivity. This hypothesis would be consistent with a wider ITCZ coinciding with reduced low-level cloud radiative effect and less positive feedbacks, and thus lower climate sensitivity; conversely, a narrower ITCZ with stronger low-level cloud radiative effects and more positive low-level cloud feedbacks would have higher climate sensitivity. This hypothesis was supported in aquaplanet experiments where tropical deep convection was varied by altering longwave cloud heating rates, with the HadGEM2-A model. With this hypothesis in mind, we return to our four pilot ITCZ-MIP models (where ITCZ width is forced by $q$-fluxes applied to slab oceans), and examine the simulations in which $CO_2$ concentrations increase from base states with differing initial ITCZ width.

ITCZ width itself does not consistently narrow with $CO_2$-driven warming in the pilot ITCZ-MIP simulations (on the $y$-axis of Fig. 10, values greater than zero indicate widening ITCZ while values less than zero indicate a narrowing ITCZ). In CESM2 and Isca simulations with narrow base climate ITCZ, the response to increased $CO_2$ is widening. In CESM2, GFDL-AM2, and Isca simulations with wide base climate ITCZ, the response to increased $CO_2$ is narrowing. For GFDL-AM2 with narrow base climate ITCZ and for all CESM1 simulations, ITCZ width stays about the same with increasing $CO_2$.

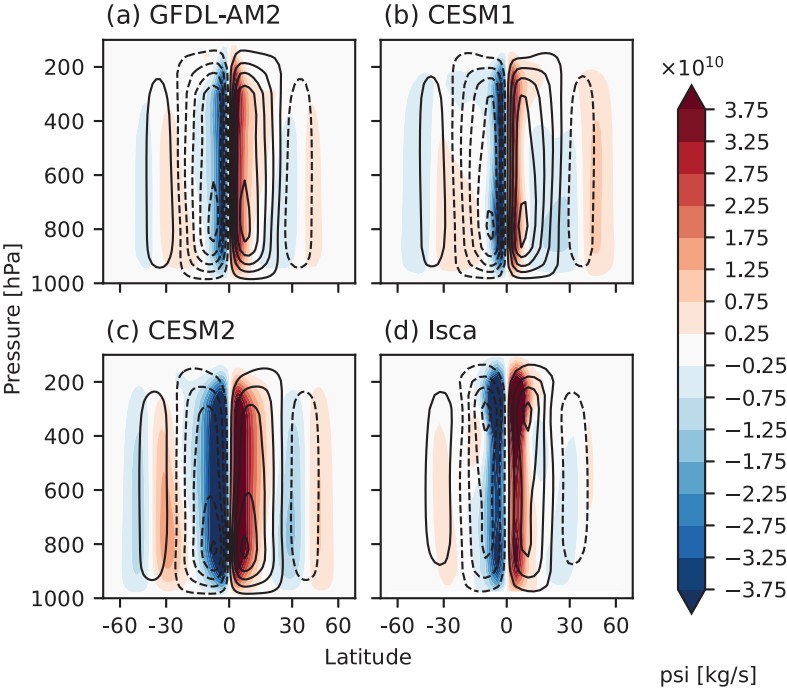

**Figure 7.** The time- and zonal-mean mass streamfunction for the slab control run (black contours in all panels) and the difference between the experiments with $A = 40\ \mathrm{W\ m^{-2}}$ and the slab control run (shading) for all four models.

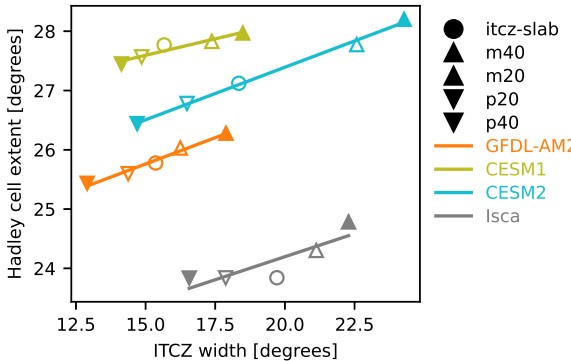

**Figure 8.** Hadley cell extent as measured by zero-crossing of the 500hPa mass streamfunction plotted against width of the ITCZ for the present-day $CO_2$ simulations of all models.

Within each model, though, there is an anti-correlation between ITCZ width in the base state and the change in ITCZ width as $CO_2$ increases (Fig. 10a). The magnitude of the relationship varies greatly among models: from very small in CESM1,

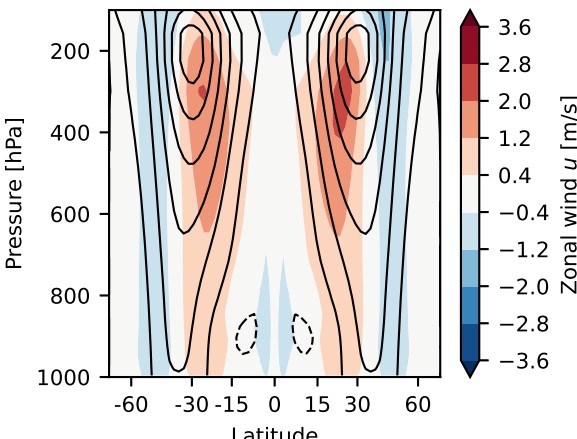

**Figure 9.** The time- and zonal-mean zonal wind for the slab control run (black contours) and the difference between the experiment with $A = 40\ \mathrm{W\ m^{-2}}$ and the slab control run (shading) for GFDL-AM2.

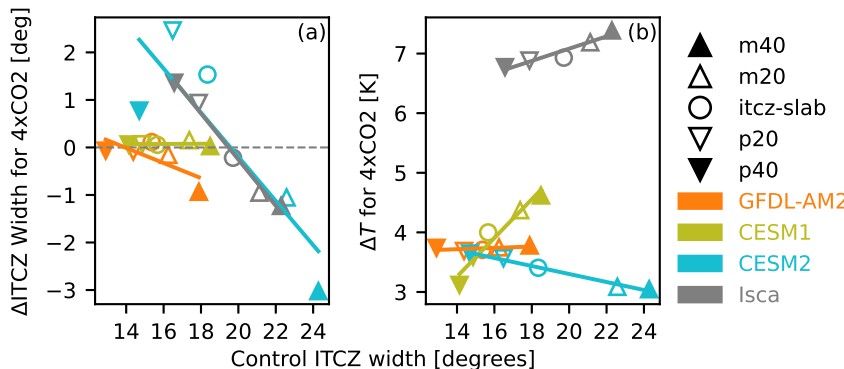

**Figure 10.** Response to an abrupt $CO_2$ quadrupling of (a) ITCZ width and (b) global mean surface temperature plotted against ITCZ width in control $CO_2$ simulations.

moderate in GFDL-AM2, to large in Isca and CESM2. Overall this suggests a preferred ITCZ width in each model, at some intermediate value, though with strongly varying degrees of affinity to the preferred width among the four models.

Previous authors have hypothesized that ITCZ width could affect climate sensitivity, and the ITCZ-MIP experiments provide an intervention with which to test this hypothesis. Among the four pilot ITCZ-MIP models, the effect on climate sensitivity of ITCZ width spans the full range of possibilities – increase, decrease, to no substantive effect (Fig. 10b). While the effect in CESM2 is consistent in direction with the expectation from Webb and Lock (2020)'s hypothesis, it is inconsistent with all other models, even including CESM1. Recalling that Isca lacks cloud radiative feedbacks, and thus we should take care interpreting

its climate sensitivity, its response is nonetheless similar to CESM1 (in terms of the slopes in Fig. 9b). So even the three pilot MIP models with cloud schemes span the full range of possible effects of ITCZ width on climate sensitivity.

     A more consistent pattern of behavior across the pilot MIP models is the relationship between the response of ITCZ width and strength, which are strongly anti-correlated in response to $CO_2$ increase (not shown). This is consistent with their relationship within the base climate state (Section 3.3) and with the response to global warming in CMIP5 (Byrne et al., 2018, Fig. 4b).

## 345    5    Discussion and Implications

The ITCZ is important for global climate, both locally in the tropics and as the source for teleconnections to the extratropical atmosphere. Yet the variability and change of its width are uncertain, not well understood, and lacking in fundamental theory. We begin to address this by developing a protocol for changing ITCZ width and demonstrating its successful application to multiple climate models. In response to added heat flux at the surface in the deep tropics, the width of the ITCZ decreases;
conversely, in response to removing surface heat flux, the ITCZ width increases. The ITCZ width response to the surface heat flux forcing is linear in sign and also magnitude.

     The ITCZ-MIP protocol for forcing ITCZ width to change is inspired by and consistent with earlier work examining differences in ITCZ width among climate models from a diagnostic energy budget perspective. In particular, the idea for perturbing ITCZ width by adding (or removing) energy from the deep tropics originated in the analysis of Byrne and Schneider (2016b).
While previous studies have generated changes in ITCZ width in individual climate models, this is the first study to use a single protocol and demonstrate a successful change in ITCZ width in more than one climate model.

     Findings from this pilot MIP prompt at least three avenues for further investigation which could be pursued by future work. First, the relationship between ITCZ width and strength is consistent across experiments and among the pilot MIP climate models, suggesting that it could be a robust feature of the ITCZ. Adequately establishing robustness among climate models
would require analyzing a larger set of climate models, as well as further examination of the possibility that it could be an artifact of the protocol. The protocol and its successful generalization to multiple climate models demonstrated here can provide a foundation on which future work can build, into a MIP with a larger number of models in order to increase ability to infer robustness.

     To demonstrate disagreement among models, on the other hand, a small set of climate models is sufficient; the pilot MIP
clearly reveals that models disagree about the relationship of climate sensitivity to base-state ITCZ width, even in sign. The second avenue, then, is to understand why. Further investigation of how ITCZ width affects climate feedbacks, especially cloud feedbacks, could delve into the mechanisms and processes underlying this disagreement.

     The third avenue for further investigation is, what, exactly, the mechanism of action is for the consistent changes in ITCZ width that are generated by the addition of heat via surface fluxes. Overall, the ITCZ-MIP protocol for generating changes
in ITCZ width reveals rich differences in behavior among models. Pursuing these avenues of further investigation – even with simple experiments run in an idealized configuration – might bear fruit in the form of fuller theories and deepened understanding of the ITCZ width and its impacts on global climate.

*Code and data availability.* Monthly 2-D model output from the pilot MIP simulations and code (in the form of python notebooks) to generate all figures and the additional $q$-flux forcing are available from https://github.com/apendergrass/width-itcz-mip (doi:10.5281/zenodo.10558340).

*Author contributions.* AGP and MPB conceptualized the project initially, MJW provided conceptual input throughout the evolution of the project, and OWM provided essential early input. MPB, AGP, and OWM led the methodology, investigation and analysis to develop the protocol. Model simulations were contributed by MPB (CESM1.2), AGP (CESM2.0), OWM (GFDL-AM2.1), and PM (Isca). OWM led the data curation and software development for analyzing the pilot MIP simulations. OWM led the initial draft of data visualizations, and AGP revised and finalized the visualizations. OWM drafted the protocol description and section 4.1 of the original manuscript draft, PM migrated 380 and reformatted content, and AGP led the writing of the rest of the original draft of the manuscript. All authors contributed to review and editing of the manuscript. AGP, MPB, OWM, and MJW responded to Reviewer comments and revised the manuscript.

*Competing interests.* The authors declare that they have no conflict of interest.

*Acknowledgements.* ITCZ-MIP was conceived at the 2017 CFMIP meeting. OWM thanks Dargan Frierson for valuable discussions regarding the experimental protocol. We thank three anonymous Reviewers and community discussant Wenyu Zhou for their constructive feedback.
AGP was supported by the U.S. Department of Energy, Office of Science, Office of Biological & Environmental Research (BER), Regional and Global Model Analysis (RGMA) component of the Earth and Environmental System Modeling Program under Award Number DE-SC0022070 and National Science Foundation (NSF) IA 1947282, and by the National Center for Atmospheric Research (NCAR), which is a major facility sponsored by the NSF under Cooperative Agreement No. 1852977, and by the Alfred P. Sloan Foundation Research Fellowship; OWM by NSF Grant AGS-1665247, the Natural Sciences and Engineering Research Council of Canada, the NOAA Climate 390 and Global Change Postdoctoral Fellowship; MJW by the Met Office Hadley Centre Climate Programme funded by DSIT; and PM by NERC (grant numbers NE/T003863/1 and NE/W005239/1).

Some of the computing resources for this project were provided by the Climate Simulation Laboratory at NCAR's Computational and Information Systems Laboratory (doi:10.5065/D6RX99HX and http://n2t.net/ark:/85065/d7wd3xhc), sponsored by the US National Science Foundation and other agencies.

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
