# Peer review of "Impact of ITCZ width on global climate: ITCZ-MIP"

_Geoscientific Model Development, 2024_

## Referee Comment (RC2)

**Review**

This study demonstrates a newly launched ITCZ-MIP. Its primary objective is to understand the width of the Intertropical Convergence Zone (ITCZ), its underlying control mechanisms, and the resulting impacts. Specifically, it provides pilot experiments to strengthen the scientific importance of the MIP. The pilot results begin with the narrowing ITCZ to the equatorial energy input, consistent with the theoretical framework and explains the relationship with the global-mean temperature. Furthermore, they report that the ITCZ width does not fully account for the proposed emergent constraint. Even though it is not totally surprising the failure as the emergent relationship would be coupled with various features behind the width of ITCZ, the proposed MIP would be valuable to dynamically assess the role of the width.

Overall, it is pleasure to read and review the manuscript. I have no doubt that this study will be published in this journal, and its scientific value will be substantial when the MIP provides various multi-model results. However, I believe that it has the potential to be further improved by revision or expansion of certain sections. Hence, I think minor revision is appropriate. As such, I have provided some comments below based on my reading of the manuscript. Please review the feedback provided for further consideration.

**Major comments**

1. Figure 1.

   1.1 I think Figure 1 should provide more information for the authors to understand both the experimental design and the basic state of the experiments. For example, the authors don't show the modified SST profile. Although they show it in Figure 4, it is not shown in Chapter 2. I think at least SST profile and q-flux profiles (both F and q_itcz) should be provided in the manuscript.

   1.2. In the current step, Figure 1 shows only q_itcz. But both caption and y_axis note it as a q-flux forcing.

2. chapter 4.1

   2.1. Why is the scatter plot not shown for Figure 7 (i.e., ITCZ width vs. Hadley cell edge)?

   2.2. CESM2 is used as a representative example for other figures; but why is Figure 8 showing GFDL-AM2?

   2.3. Chapter 4.2 explains the background hypothesis well, which helps to understand the pilot results. However, I carefully argue that it looks a bit rushed in chapter 4.1. Let's assume that the narrowing of the ITCZ is accompanied by a strengthening of the Hadley cell. The upper tropospheric momentum transport would be enhanced, resulting in the

acceleration of the subtropical jet. Previous studies consistently show that the enhanced subtropical jet provides a fertile region for baroclinicity, leading to an equatorward eddy-driven jet (e.g., Lee and Kim 2003; Brayshaw et al. 2008; Shin and Kang 2021). I think this mechanism would be a plausible explanation, or at least a possible hypothesis, in Section 4.1. Whatever the connection, the potential explanation for the effect should be mentioned in the manuscript.

**Minor comments**

1. "These results indicate that idealized model experiments have the potential to increase our understanding of ITCZ width."(L8) would be too ambiguous and general implication. I think it would be better to represent meaningful insights already shown in pilot results and/or what we could learn from when the MIP is fully prepared.

2. Introduction. The authors well explain the global effects of the ITCZ (and its width) in the introduction. However, I find the regional aspects lacking. In particular, the width of the ITCZ is closely related to the hydrological cycle over tropical countries, given its sharp meridional structure. The regional aspects may be typical and natural, but they are still worth mentioning.

3. L17-29. I think these paragraphs should be reorganized. Particularly, L19-21 tells the potential impact of the ITCZ width, whereas the proposed mechanism present from L22. Accompanies with previous comments, intuitive importance could be easily shown in regional aspects.

4. L51. Is the lack of a seasonal cycle a requirement for the aquaplanet configuration? Personally, I don't agree with this. Of course, it is natural to exclude seasonality in idealized experiments. At the same time, there are a number of aquaplanet experiments that take advantage of the seasonal cycle (e.g., Kim et al. 2018; Feldl and Merlis 2021).

5. L57. I think there are more experiments which utilize multi-model aquaplanet experiments (e.g., Seo et al. 2017; Voigt et al. 2016)

6. L62. "Most simulations have a slab ocean with a 10-m mixed layer depth." As the itcz-SST is only for the reproducing SST profile, I think this would make confusion on the experimental design.

7. L73-77. It would be great if the authors could give a more detailed reason for a single ITCZ. This is also related with the ITCZ metric. Because the author's ITCZ metric could be appliable for double-ITCZ, more detail reason would be helpful.

8. L85-89. I think this paragraph should be shown in earlier (e.g. L53-54 where explains the slab ocean). For me, the paragraph hesitates to understand the overall experimental design, especially the role of itcz-SST experiments.

9. Figure 3. Please include definition of the line in the caption.

10. Figure 4. I think the relationship would be related to the Qflux structure that could accompany with the non-linear SWCRE. Of course, this is somewhat in-line with second potential reason. It is supported by not only the downward TOA flux is dominated by the SW, but also the GFDL-AM2 shows the largest sensitivity of global-mean temperature (the model is already known for non-linear SWCRE; e.g., Shaw et al. 2015; Shin et al. 2017). I think this possibility is also worth noting.

11. L262. "And why does the multi-model mean response (Byrne et al., 2018) differ?" I think it would be easier to explain more how multi-model mean response here.

12. Momentum Perspective. I am just wondering what the authors think about the use of momentum (e.g. the rotation of the planet) compared to the imposing Qflux. As presented by the authors, the imposing Qflux has the potential to disrupt the climatology without adding additional heat. Of course, if the planet's rotation is changed, not only the general circulation but also the thermal structure would be consequently changed. But still, in terms of additional (or artificial) heating, it has advantages. I'm not saying that the MIP should be reconsidered/organized with the momentum perspective, but it's still worth noting the expert opinion in the discussion section.

Reference

Brayshaw, D. J., B. Hoskins, and M. Blackburn, 2008: The Storm-Track Response to Idealized SST Perturbations in an Aquaplanet GCM. *J. Atmospheric Sci.*, **65**, 2842–2860, https://doi.org/10.1175/2008JAS2657.1.

Feldl, N., and T. M. Merlis, 2021: Polar Amplification in Idealized Climates: The Role of Ice, Moisture, and Seasons. *Geophys. Res. Lett.*, **48**, e2021GL094130, https://doi.org/10.1029/2021GL094130.

Kim, D., S. M. Kang, Y. Shin, and N. Feldl, 2018: Sensitivity of Polar Amplification to Varying Insolation Conditions. *J. Clim.*, **31**, 4933–4947, https://doi.org/10.1175/JCLI-D-17-0627.1.

Lee, S., and H. Kim, 2003: The Dynamical Relationship between Subtropical and Eddy-Driven Jets. *J. Atmospheric Sci.*, **60**, 1490–1503, https://doi.org/10.1175/1520-0469(2003)060<1490:TDRBSA>2.0.CO;2.

Seo, J., S. M. Kang, and T. M. Merlis, 2017: A model intercomparison of the tropical precipitation response to a CO2 doubling in aquaplanet simulations. *Geophys. Res. Lett.*, **44**, 993–1000, https://doi.org/10.1002/2016GL072347.

Shaw, T. A., A. Voigt, S. M. Kang, and J. Seo, 2015: Response of the intertropical convergence zone to zonally asymmetric subtropical surface forcings. *Geophys. Res. Lett.*, **42**, 9961–9969, https://doi.org/10.1002/2015GL066027.

Shin, Y., and S. M. Kang, 2021: How Does the High-Latitude Thermal Forcing in One Hemisphere Affect the Other Hemisphere? *Geophys. Res. Lett.*, **48**, e2021GL095870, https://doi.org/10.1029/2021GL095870.

——, ——, and M. Watanabe, 2017: Dependence of Arctic climate on the latitudinal position of stationary waves and to high-latitudes surface warming. *Clim. Dyn.*, **49**, 3753–3763, https://doi.org/10.1007/s00382-017-3543-y.

Voigt, A., and Coauthors, 2016: The tropical rain belts with an annual cycle and a continent model intercomparison project: TRACMIP. *J. Adv. Model. Earth Syst.*, **8**, 1868–1891, https://doi.org/10.1002/2016MS000748.

---

## Author Comment (AC1)

**Response to Reviewer 3 comments**

*[Review comments in Heading 6 style*

Our responses in Normal text style]

*This manuscript outlines the protocol for a proposed model intercomparison project (MIP) to study the Intertropical Convergence Zone (ITCZ). The proposed ITCZ-MIP has a series of experiments (separated into two tiers) as well as a concrete list of requested output from each simulation. The other portion of the manuscript is a pilot study using four models that run the ITCZ-MIP experiments. The authors demonstrate their methodology achieves the desired result of predictably altering the ITCZ width in their simulations. Their methodology does, however, create a significant change in global mean temperature, despite imposing zero global surface flux, which they show to result from changes in outgoing longwave radiation. The final portion of the manuscript covers the role of the ITCZ in a changing climate. Here, the models are far less consistent, and the authors use this inconsistency to motivate a need for more models to contribute to ITCZ-MIP.*

*My primary concern has to do with how this study relates to the previous tropical rain belts with an annual cycle and a continent model intercomparison project (TRACMIP). The experiment design of TRACMIP was built on the premise that the seasonality is important to understanding the ITCZ. Specifically, TRACMIP was meant to address the finding that sensitivity of the ITCZ to model changes in equinoctial aquaplanet simulations can be very different from simulations with a seasonal cycle (see Figure 1 of Voigt et al., 2016). I worry that by using equinoctial conditions for this study, the methods and conclusions drawn from ITCZ-MIP might have limited applicability to the ITCZ biases commonly found in fully coupled earth system models. Similarly, Zhou et al. (2020) showed that changes in the annual mean ITCZ width can be complicated by its seasonal nature. They found the ITCZ changes both its position and width within each season in such a way that the annual mean ITCZ appears to narrow, when the ITCZ actually widens at any time within the seasonal cycle. While there are opportunities for scientific understanding with simpler experiment designs, I would have liked to have seen more discussion about the potential limitations of the experimental design considering the lack of seasonality.*

An important motivation we had for not including the seasonal cycle, which this comment made us realize we had not stated in the submitted manuscript, were observational findings of the author team. In the zonal mean over ocean, the seasonal cycle of the distribution of daily rain amount (as a function of daily intensity) is largely driven by the frequency of precipitation within the rain belts, rather than meridional migration of the rain belts, which dominates over land

(Pendergrass and Deser 2017). We will also include this motivation and citation in the introduction.

Looking back at Fig 1 of Voigt et al., (2016), this figure isn't emphasizing the seasonality in isolation or alone; but rather, seasonality is added along with a slab ocean that allows SST to freely evolve. We fully agree that prescribed SSTs are too simple for understanding ITCZ width; that's why our protocol focuses on slab ocean simulations (the one prescribed-SST simulation is just a necessary technical step to be able to run the slab ocean). We also agree that for studying ITCZ position and continental rainfall (the stated purposes of TRACMIP, in the abstract of Voigt et al., 2016), a seasonal cycle is appropriate.

Part of the knowledge gap we were responding to at the time was something we thought TRACMIP was not designed to look at, and few others (again, in 2017) were looking at, was ITCZ width. For a focus specific to ITCZ width, we thought that including the seasonal cycle could introduce a rectification problem, while excluding the seasonal cycle would help narrow the focus to the physical rain belt itself, separating the width of the rain belt from its movement over the course of the year - perhaps obscuring elements of the ITCZ width that might be important.

Zhou et al. (2020) certainly emphasize that averaging over multiple seasons complicates the picture of the ITCZ. They address this by analyzing individual seasons and basin separately, which is really all that can be done in a comprehensive configuration with continents and a seasonal cycle. Since we are not looking at the rectification of the seasonal cycle to get the annual mean, but rather simply removing the seasonal migration altogether, we think their findings actually provide motivation for removing the seasonal cycle - to get at the essence of the problem of what a band of precipitation does. Our configuration is better thought of as a single perpetual season, rather than an annual mean.

Furthermore, their complex and comprehensive configuration means that in the simulations Zhou et al., (2020) analyze, there is a potential for interaction between the location and the width of the ITCZ within individual seasons. In contrast, our equatorially-symmetric design, with no seasonal migration, removes the potential for this interactions between ITCZ width and location.

Overall, our goal with the ITCZ-MIP experimental design was to remove as many complexities as possible while still retaining the minimum necessary for ITCZ width to vary. The inclusion of a seasonal cycle would add complexity. Given the relative lack of research on ITCZ width and its role in climate, it seems sensible to start this MIP with a very idealised setup (i.e. no seasonal cycle) and understand that before adding complexity down the line (as other MIPs have done, RCEMIP being a recent example; Wing et al. 2018).

In the text, will add a paragraph about the exclusion of the seasonal cycle citing Zhou et al., (2020) and Pendergrass and Deser (2017), as well as Voigt et al., (2016; though this last paper was already cited in the original submission).

*While the above concern is one I believe requires a major revision to address fully, the rest of the manuscript is well written and suitable for publication in GMD. I have a few additional minor comments below.*

*Minor comments*

*L82 "This narrower peak in SST…" should this reference Figure 4?*

No, we didn't include comparisons between the Control SST and Qobs profiles in the manuscript, since that's already documented in the literature (cited in the sentence). To prevent readers from making this incorrect connection, we'll explicitly state "not shown here".

*L298 "…its response is nonetheless similar to CESM1." Did you mean CESM2?*

No, we meant CESM1. I think your confusion might have been that Isca and CESM2 are extremely similar in Fig. 9a (which shows the change in ITCZ width for 4xCO2). What we were trying to refer to was the slope of the lines in Fig. 9b, in which Isca has the same sign as CESM1, but the opposite sign from CESM2. Visually they don't look as similar, because of the offset in control temperature change, which we presume is due to Isca's idealized radiation.

To try and make this clearer for readers, we'll add " (in terms of the slopes in Fig.~9b)" to the end of this sentence.

*L301 "This is consistent with the their relationship…" Typo*

Thanks for pointing out this typo. We will remove "the" in the updated draft.

**Additional References**

Pendergrass, A. G., and C. Deser, 2017: Climatological characteristics of typical daily precipitation. *Journal of Climate*, **30**, 5985–6003, https://doi.org/10.1175/JCLI-D-16-0684.1.

Wing, A. A., K. A. Reed, M. Satoh, B. Stevens, S. Bony, and T. Ohno, 2018: Radiative–convective equilibrium model intercomparison project. *Geoscientific Model Development*, **11**, 793–813, https://doi.org/10.5194/gmd-11-793-2018.

Zhou, W., L. R. Leung, J. Lu, D. Yang, and F. Song, 2020: Contrasting Recent and Future ITCZ Changes From Distinct Tropical Warming Patterns. *Geophysical Research Letters*, **47**, e2020GL089846, https://doi.org/10.1029/2020GL089846.

---

## Author Comment (AC2)

**Response to comments from Anonymous Reviewer 2**

[Review comments in gray italic style

Our responses in plain text style]

Review

This study demonstrates a newly launched ITCZ-MIP. Its primary objective is to understand the width of the Intertropical Convergence Zone (ITCZ), its underlying control mechanisms, and the resulting impacts. Specifically, it provides pilot experiments to strengthen the scientific importance of the MIP. The pilot results begin with the narrowing ITCZ to the equatorial energy input, consistent with the theoretical framework and explains the relationship with the global-mean temperature. Furthermore, they report that the ITCZ width does not fully account for the proposed emergent constraint. Even though it is not totally surprising the failure as the emergent relationship would be coupled with various features behind the width of ITCZ, the proposed MIP would be valuable to dynamically assess the role of the width.

Overall, it is pleasure to read and review the manuscript. I have no doubt that this study will be published in this journal, and its scientific value will be substantial when the MIP provides various multi-model results. However, I believe that it has the potential to be further improved by revision or expansion of certain sections. Hence, I think minor revision is appropriate. As such, I have provided some comments below based on my reading of the manuscript. Please review the feedback provided for further consideration.

**Major comments**

**1. Figure 1.**

1.1 I think Figure 1 should provide more information for the authors to understand both the experimental design and the basic state of the experiments. For example, the authors don't show the modified SST profile. Although they show it in Figure 4, it is not shown in

2

Chapter 2. I think at least SST profile and q-flux profiles (both F and q\_itcz) should be

provided in the manuscript.

The novel component of the experimental design documented in this manuscript is the intervention of additional heat fluxes that changes the ITCZ width. We chose Fig. 1 because of this, and we think adding base state information to this particular figure would dilute that purpose.

The baseline heat fluxes needed to drive a slab ocean are standard elements of any slab ocean configuration; they are not new elements we are introducing here. Furthermore, they are specific to each model - someone running simulations following the protocol would need to generate these themselves.

That said, we understand that they may be useful in interpreting and understanding the changes in ITCZ width in the simulations, though we haven't gotten to that level of depth in this manuscript (whose scope is limited to documenting the experimental design). To address this, we will add a supplementary figure documenting the baseline heat fluxes for each pilot model's itcz-slab experiment and the resulting SST climatology.

1.2. In the current step, Figure 1 shows only q\_itcz. But both caption and y\_axis note it as

**a q-flux forcing.**

Thanks for pointing this out. In terms of the calculation it is an *additional* heat flux, added to the model-specific standard heat flux (F in equations 2 and 5). We will change the y-axis and caption. Additionally, we will change it in the title of subsection 2.3.

**2. chapter 4.1**

2.1. Why is the scatter plot not shown for Figure 7 (i.e., ITCZ width vs. Hadley cell edge)?

We tried hard to keep the original manuscript focused on the problem at hand; perhaps our focus was too narrow on this point. We will add the scatterplot of ITCZ width versus Hadley cell extent in the updated manuscript.

2.2. CESM2 is used as a representative example for other figures; but why is Figure 8

**showing GFDL-AM2?**

In the cases where we show just one model's output rather than all 4, we chose to do this in order to focus the reader's attention on the experimental design and its effects by showing an

example of model output; including plots for all four models would distract readers with the differences among models, which in these cases were not the reason we were showing the plots. The sections of the paper that author Pendergrass led use CESM2 as the example model, because she ran the CESM2 simulations. The sections of the paper that author Watt-Meyer wrote use GFDL-AM2, because he ran the GFDL-AM2 simulations. Author contributions in terms of model simulations (who ran which model) and sections of the paper led are documented in the Author Contribution statement in the manuscript.

**2.3. Chapter 4.2 explains the background hypothesis well, which helps to understand the**

pilot results. However, I carefully argue that it looks a bit rushed in chapter 4.1. Let's

assume that the narrowing of the ITCZ is accompanied by a strengthening of the Hadley

cell. The upper tropospheric momentum transport would be enhanced, resulting in the acceleration of the subtropical jet. Previous studies consistently show that the enhanced

subtropical jet provides a fertile region for baroclinicity, leading to an equatorward eddydriven

jet (e.g., Lee and Kim 2003; Brayshaw et al. 2008; Shin and Kang 2021). I think

this mechanism would be a plausible explanation, or at least a possible hypothesis, in

Section 4.1. Whatever the connection, the potential explanation for the effect should be

mentioned in the manuscript.

Thanks for the comment. We generally agree with this interpretation and added the following text to this section: "These circulation responses, which coincide with a strengthening of the subtropical jet (Fig. 9 [Fig. 8 of original draft]) could be explained as follows. A narrower ITCZ implies greater poleward transport of zonal angular momentum in the upper troposphere, and hence a stronger subtropical jet. The stronger subtropical jet in turn leads to increased baroclinicity at lower latitudes and therefore an equatorward shifted Hadley cell edge and eddy-driven jet (Lee and Kim, 2003; Brayshaw et al., 2008; Watt-Meyer and Frierson, 2019)."

**Minor comments**

1. "These results indicate that idealized model experiments have the potential to increase our understanding of ITCZ width."(L8) would be too ambiguous and general implication. I think it would be better to represent meaningful insights already shown in pilot results and/or what we could learn from when the MIP is fully prepared.

**As suggested by the reviewer, we have replaced the final sentence of the abstract with two new sentences that: (i) articulate the insights into ITCZ width gained from the pilot study; and (ii) highlight the advances expected from a comprehensive MIP focused on ITCZ width.**

2. Introduction. The authors well explain the global effects of the ITCZ (and its width) in the

introduction. However, I find the regional aspects lacking. In particular, the width of the

ITCZ is closely related to the hydrological cycle over tropical countries, given its sharp

meridional structure. The regional aspects may be typical and natural, but they are still worth

**mentioning.**

Thanks for pointing this out. We certainly appreciate wanting to do work that's relevant. Since our main goal with this project is to strip the ITCZ back to its essence, and we thought that the knowledge gap that might be able to be filled here was the radically idealized end of the spectrum where zonal asymmetries and land are removed from the experimental design, we are wary of misdirecting readers by indicating that we might address specifics of precipitation over tropical land. But as a long-term outcome downstream from the project, we would of course expect that better understanding of the essence of the ITCZ width would eventually also be useful in realistic settings.

We've added a new sentence to the first paragraph of the introduction: "The ITCZ dominates regional hydroclimate in tropical regions, which are vulnerable to even modest changes in its width, strength, or position."

3. L17-29. I think these paragraphs should be reorganized. Particularly, L19-21 tells the

potential impact of the ITCZ width, whereas the proposed mechanism present from L22.

Accompanies with previous comments, intuitive importance could be easily shown in

regional aspects.

The first two paragraphs of the introduction have been re-organised to more clearly separate discussions of: (i) the potential impacts of ITCZ width on broader climate; and (ii) the processes controlling ITCZ width.

4. L51. Is the lack of a seasonal cycle a requirement for the aquaplanet configuration?

Personally, I don't agree with this. Of course, it is natural to exclude seasonality in idealized

experiments. At the same time, there are a number of aquaplanet experiments that take

advantage of the seasonal cycle (e.g., Kim et al. 2018; Feldl and Merlis 2021).

Thanks for pointing out this oversight. Yes it's possible for aquaplanets to include the seasonal cycle; we should have stated that we chose not to include it. We will revise this statement accordingly.

5. L57. I think there are more experiments which utilize multi-model aquaplanet experiments

(e.g., Seo et al. 2017; Voigt et al. 2016)

We cited Stevens and Bony (2013) here as a reference for the statement, "tropical clouds and precipitation have dramatically different responses to increasing greenhouse gases in different climate models, even in simulations with an aquaplanet"; this point was the main thesis of this particular paper (which was a perspective rather than a standard research article). There are plenty of MIPs using aquaplanets, but it's not obvious to us how adding such a list would advance the line of reasoning we're trying to bring the reader through.

Since this was the first use of the term "aquaplanet," we also defined the term in this sentence, and it ended up in between the statement the citation was for and the citation itself; perhaps this led to confusion. We've moved the citation closer to the relevant statement to address this.

6. L62. "Most simulations have a slab ocean with a 10-m mixed layer depth." As the itcz-SST

is only for the reproducing SST profile, I think this would make confusion on the

experimental design.

We have added text to the referenced sentence to flag the prescribed-SST simulation and that it is described in the subsequent subsection.

7. L73-77. It would be great if the authors could give a more detailed reason for a single

ITCZ. This is also related with the ITCZ metric. Because the author's ITCZ metric could be

appliable for double-ITCZ, more detail reason would be helpful.

In response to this comment we have modified the sentences referenced by the reviewer. In particular, in the modifications our aim is to highlight that although the double-ITCZ problem is an interesting one, the processes controlling the width of a single ITCZ remain debated as does the influence of a single ITCZ on broader climate. So, in our view, it is important to tackle the "simpler" single-ITCZ problem in its own right. There is a considerable body of research on the double-ITCZ problem, which we think our study will neatly complement.

8. L85-89. I think this paragraph should be shown in earlier (e.g. L53-54 where explains the

slab ocean). For me, the paragraph hesitates to understand the overall experimental design,

especially the role of itcz-SST experiments.

We will make this change in the updated manuscript.

9. Figure 3. Please include definition of the line in the caption.

Thanks for pointing out this oversight. We will add it in the revised version.

10. Figure 4. I think the relationship would be related to the Qflux structure that could accompany with the non-linear SWCRE. Of course, this is somewhat in-line with second potential reason. It is supported by not only the downward TOA flux is dominated by the SW, but also the GFDL-AM2 shows the largest sensitivity of global-mean temperature (the model is already known for non-linear SWCRE; e.g., Shaw et al. 2015; Shin et al. 2017). I think this possibility is also worth noting.

**Thank you for pointing this out. We will include a new paragraph noting this in the updated manuscript.**

11. L262. "And why does the multi-model mean response (Byrne et al., 2018) differ?" I think

**it would be easier to explain more how multi-model mean response here.**

The first paragraph of this section (3.3) describes the response of the ITCZ width to global warming in CMIP5 models. The goal of placing this question here was rhetorical: to exit the section at the same level as we started. We don't want to over-emphasize, though, what we think there is to learn from analyzing the multi-model mean in this case, since the variations in ITCZ strength across models are very large (in both directions and in sign; see Byrne et al. 2018 Fig 4b).

Instead, we will omit this sentence from the end of the section, since the important points about the (CMIP) climate model relationship between changes in ITCZ strength and width are already stated in the first paragraph of this section.

12. Momentum Perspective. I am just wondering what the authors think about the use of momentum (e.g. the rotation of the planet) compared to the imposing Qflux. As presented by the authors, the imposing Qflux has the potential to disrupt the climatology without adding additional heat. Of course, if the planet's rotation is changed, not only the general circulation but also the thermal structure would be consequently changed. But still, in terms of additional (or artificial) heating, it has advantages. I'm not saying that the MIP should be reconsidered/organized with the momentum perspective, but it's still worth noting the expert

**opinion in the discussion section.**

Interesting idea! We're glad if our manuscript is sparking new ideas like this, which are potentially interesting though another MIP or an extension of this MIP with a new focus (on momentum) and new experimental design, e.g., including simulations with variable rotation rates. Though interesting, we think this is distinct enough from and outside the scope of the current work that this kind of commentary would distract from the current manuscript more than it would contribute.

Reference

Brayshaw, D. J., B. Hoskins, and M. Blackburn, 2008: The Storm-Track Response to Idealized SST

Perturbations in an Aquaplanet GCM. J. Atmospheric Sci., 65, 2842–2860,

https://doi.org/10.1175/2008JAS2657.1.

*Feldl, N., and T. M. Merlis, 2021: Polar Amplification in Idealized Climates: The Role of Ice, Moisture,*

and Seasons. Geophys. Res. Lett., 48, e2021GL094130, https://doi.org/10.1029/2021GL094130.

*Kim, D., S. M. Kang, Y. Shin, and N. Feldl, 2018: Sensitivity of Polar Amplification to Varying Insolation*

Conditions. J. Clim., 31, 4933–4947, https://doi.org/10.1175/JCLI-D-17-0627.1.

Lee, S., and H. Kim, 2003: The Dynamical Relationship between Subtropical and Eddy-Driven Jets. J.

Atmospheric Sci., 60, 1490–1503, https://doi.org/10.1175/1520-

0469(2003)060<1490:TDRBSA>2.0.CO;2.

Seo, J., S. M. Kang, and T. M. Merlis, 2017: A model intercomparison of the tropical precipitation

response to a CO2 doubling in aquaplanet simulations. Geophys. Res. Lett., 44, 993–1000,

https://doi.org/10.1002/2016GL072347.

Shaw, T. A., A. Voigt, S. M. Kang, and J. Seo, 2015: Response of the intertropical convergence zone to

zonally asymmetric subtropical surface forcings. Geophys. Res. Lett., 42, 9961–9969,

https://doi.org/10.1002/2015GL066027.

Shin, Y., and S. M. Kang, 2021: How Does the High-Latitude Thermal Forcing in One Hemisphere Affect

the Other Hemisphere? Geophys. Res. Lett., 48, e2021GL095870,

https://doi.org/10.1029/2021GL095870.

——, ——, and M. Watanabe, 2017: Dependence of Arctic climate on the latitudinal position of stationary

waves and to high-latitudes surface warming. Clim. Dyn., 49, 3753–3763,

https://doi.org/10.1007/s00382-017-3543-y.

*Voigt, A., and Coauthors, 2016: The tropical rain belts with an annual cycle and a continent model*

intercomparison project: TRACMIP. J. Adv. Model. Earth Syst., 8, 1868–1891,

https://doi.org/10.1002/2016MS000748.

**Additional References**

- Donohoe, A., D. M. W. Frierson, and D. S. Battisti, 2014: The effect of ocean mixed layer depth on climate in slab ocean aquaplanet experiments. *Climate Dynamics*, **43**, 1041–1055, https://doi.org/10.1007/s00382-013-1843-4.
- Pendergrass, A. G., and C. Deser, 2017: Climatological characteristics of typical daily precipitation. *Journal of Climate*, **30**, 5985–6003, https://doi.org/10.1175/JCLI-D-16-0684.1.

Wing, A. A., K. A. Reed, M. Satoh, B. Stevens, S. Bony, and T. Ohno, 2018: Radiative–convective equilibrium model intercomparison project. *Geoscientific Model Development*, **11**, 793–813, https://doi.org/10.5194/gmd-11-793-2018.

- Zhou, W., S.-P. Xie, and D. Yang, 2019: Enhanced equatorial warming causes deep-tropical contraction and subtropical monsoon shift. *Nature Climate Change*, https://doi.org/10.1038/s41558-019-0603-9.
- —, L. R. Leung, J. Lu, D. Yang, and F. Song, 2020: Contrasting Recent and Future ITCZ Changes From Distinct Tropical Warming Patterns. *Geophysical Research Letters*, **47**, e2020GL089846, https://doi.org/10.1029/2020GL089846.

---

## Author Comment (AC3)

**Response to comments from Anonymous Reviewer 1**

*[Reviewer comments in gray italic style*

Our responses in Normal text style]

*The "Impact of ITCA width on global climate: ITCZ-MIP" outlines an experimental protocol for a MIP, and presents some preliminary statistics of circulation shifts, ITCZ width and strength changes, and radiative budgets. The authors then pose three questions which the MIP seems well-suited to address.*

*I think this paper is a fantastic opening discussion piece for the ITCZ-MIP, that the writing is mostly-well-organized, easy to follow, and well-cited, and that the figure production is exceptionally clean and consistent. I think if I were to contribute to the MIP, this paper would give me a good template to follow. I recommend publication pending minor revisions. I have a couple of minor points about scientific clarity and reproducibility and some similarly minor points about the text itself.*

*# Questions/comments/suggestions*

*I've been around some of the discussion of this MIP as a possibility, but I haven't been active in this community lately. So my main question may be one that is obvious to anyone focusing on ITCZ widening: what is it about the proposed and prototyped experimental protocol in particular that makes it well-suited in the authors' opinion to shedding light on the mechanisms of ITCZ? The introduction mentioned work by co-author Byrne with Schneider and others, and it sounded like the MIP was going to address possible mechanisms for ITCZ width – such as they are. The two questions posed on lines 40–45. But it seems like it will do so indirectly; most of the discussion concerns inter-model differences, etc., rather than by directly addressing the relevance of proposed mechanisms.*

Our initial ambition, that catalyzed the project, was to come up with an experimental design that would both allow us to study the mechanism for how the ITCZ width varied and also look at its effects on features important to global climate like the subtropical cloud decks that drive shortwave cloud feedbacks, Hadley cell width which intuitively seems likely to relate to ITCZ width, extratropical circulation, and the hydrologic cycle. We originally set out to design a MIP

because those have been useful for related topics, like the ITCZ location, tropical-extratropical interactions, among many others. Also, this seemed to be a good model for a group of early career scientists to be able to collaborate together, each contributing from different models they (we) were familiar with using.

Actually coming up with a protocol that works, and in multiple models, turns out to be challenging. We found that it is hard to come up with an experimental design fit for getting at all of the questions we initially hoped to be able to answer. What we did come up with, though (described here), we think is fit for the purpose of answering some questions that are interesting - specifically, the effect of ITCZ width and its variations on global climate.

We are less certain that the experimental design described here will allow us to explore the mechanisms that control the ITCZ width itself as well as we would like. But we do think it might still be useful for shedding a little more light on the mechanisms, somewhat indirectly, especially further exploring the strong relationship between ITCZ width and strength.

In considering your comment, we realized that the first sentence of the abstract probably arose more from our initial ambitions than from what we have actually accomplished and described in the manuscript. We think revising this sentence in particular will go a long way to preventing this misdirection; it had already been excised out of the introduction.

*While the protocol seems to get good bang for the buck as far as strongly shifting the ITCZ while maintaining a recognizably earth-like climate, I can't think of anything in the text that explains why the protocol is precisely what it is—or what alternatives were considered (and ultimately rejected for the prototype).*

We tried a few different approaches, inspired by the literature, including adding heat fluxes directly to the atmosphere (rather than via a slab ocean), and we considered a few different ways of doing this inspired by various of the literature we cite in the manuscript. One of the notions we worked from is that anything that was hard for any one of us to do would not be very likely to be successful as a MIP, in terms of making it easy and therefore more feasible and likely for modeling groups to run. So we did not push particular hard on methods that couldn't be readily implemented in one of our models, or methods that didn't change the ITCZ when we thought they might.

So while we are confident that this method does vary the ITCZ width, it is probably not the only way to achieve the outcome, it is just the first one that worked in the models we were familiar with.

*It would have been nice to say something about how sensitive the ITCZ is to heat fluxes,*

The experimental design has multiple amplitudes of forcing (20 and 40 W/m2 at the equator) specifically so the sensitivity to magnitude of heat fluxes can be studied. It's extremely linear in the range we tested. It is much more linear than we expected, which we tried to state in the manuscript. When behavior is so linear, it doesn't leave a lot else to say. Effects are half as big with 20 W/m2 peak forcing as with 40.

We also tested the sign of the forcing, and were also surprised at how linear that is.

This is all reflected by the excellent fit of the linear regression lines to each model shown on the scatter plots, and is described in the text.

*and how a MIP design that forced ITCZ movement some other way may be less effective because of damping from SST changes (in a coupled ocean simulation). Q-fluxes seem to be the way to go, but I recommend explaining this more clearly in the manuscript.*

In a slab ocean configuration, SSTs freely evolve, so interactions between the ITCZ and SSTs are also captured in these simulations, which focus around the slab ocean. All the experiments here are run in slab ocean, with the exception of itcz-SST, whose sole purpose is to calculating the forcing for slab ocean simulations - it is not intended for interpretation or analysis.

*I think the discussion of the confounding effect of global mean temperature change was very good.*

*It's interesting to me that the MIP is named ITCZ-MIP, but it is so squarely focused on ITCZ width and strength, when there are other important aspects that could be addressed in an ITCZ-MIP. For example, the protocol in the submission is intentionally designed to keep the ITCZ at the equator, so another ITCZ-MIP might be necessary to analyze changes in ITCZ location. That said, the ITCZ location has been the subject of considerable work lately, so I don't see the same urgent need to focus on it. But maybe this is really an ITCZW-MIP?*

Yes, this MIP is focused on ITCZ width, and does not aim to address every aspect of the ITCZ and its changes. Every time we have tried to change the name, we have failed to come up with anything else that sticks, and we go back to calling it ITCZ MIP. Since this name, while not perfect, is already in use, we prefer to leave it as it is.

*Line 17: "width of the ITCZ and its changes" is syntactically ambiguous. Context helps here, but maybe re-word this sentence.*

Thank you. We will change this to "width of the ITCZ and its changes in width" which we think resolves the ambiguity.

*Line 26: "since" suggests that the following cited papers would be after Byrne et al. (2018), but Harrop and Hartmann (2016) is obviously not. I wonder if the citation on line 27 should be written as (Dixit et al., 2018; see also Harrop and Hartmann, 2016).*

Thanks for this suggestion. This was awkward to articulate because Byrne et al (2018) talked about Harrop and Hartmann (2016)'s theory, but Dixit et al (2018) was an important update subsequent to the writing up Byrne et al. (2018). This is a good solution and we will implement it.

*Line 63: "Most simulations have a slab ocean with a 10-m mixed layer depth." I wonder whether more needs to be said about this. There's no seasonal cycle, so maybe it's not an issue. Is this something that future MIP contributors need to get right?*

The importance of slab ocean depth in aquaplanets (with a seasonal cycle) was discussed by (Donohoe et al. 2014); our understanding is that in the middle range of slab ocean depths (like 10 m), the slab ocean depth should hopefully not affect the behavior qualitatively. Of practical relevance, a system with higher heat capacity will take longer to spin up. Basically, we don't expect it to be the most important factor in the protocol setup, but we thought we should document some a default choice of it for the purpose of the protocol. If MIP contributors had 8 or 12 m depth, it seems like that shouldn't be cause for concern. On the other hand, if a group were to choose 2 or 50 m instead of 10 m, then based on previous work we might expect to see qualitative changes in behavior of the system. And practically, for 50 m more spin up time would need to be excluded from the beginning of simulations.

To reflect this in the text, without dwelling on it excessively since we don't expect it to be of particular importance, we will add a brief reference to Donohoe et al., (2014) and a mention of the expected practical impact on spin up time.

*Lines 42–45, 315–330: There are five big questions in this paper. The first two, which I call A1 and A2, are said to be addressed by this paper:*

*A1 - What mechanisms connect the ITCZ width to the width of the Hadley cell and the location of the storm tracks?*

*A2 – does equilibrium climate sensitivity depend on the ITCZ width?*

*A1 is addressed a little bit, as the Webb and Lock (2020) hypothesis is not rejected. But no alternative hypothesis is suggested.*

First addressing the comments on these two points, which were found in Lines 42-45 of the original manuscript.

The paragraph referred to by the reviewer is the final paragraph of the introduction. It is common in our field to provide an outline of the rest of the manuscript in this paragraph, which is what we were trying to do with it. So, in it, we described the sections contained in the manuscript. This manuscript has been submitted as a "model experiment description" paper; consistent with this, the main contribution is not to answer any big questions, but rather to document a model experiment. The big questions that motivate the project are not addressed in this manuscript. We do make some comments on small questions that are straightforward enough to be addressed without too much text (e.g. the effect on the Hadley cell and storm track locations).

As far as climate sensitivity, we thought it was interesting that these 4 models have differences in sign. And since the model output presented here made it possible to comment on the Webb and Lock (2020) hypothesis, a segment of the authors team was eager to do so. [That hypothesis is very relevant to climate sensitivity and not very relevant to the Hadley cell width, so we will respond on the assumption that Reviewer meant A2 instead of A1 in reference to Webb and Locke 2020.] In order to properly test the hypothesis, more analysis (e.g., full climate feedback analysis applying radiative kernels and/or cloud kernels) is required; this is beyond the scope of this model experiment description paper, and one among multiple big questions the overall project aspires to address.

The main point of the project is to understand the ITCZ width and its variations, and how they affect climate. The main point of this manuscript is to document a set of model experiments: describe why we think they might be useful, what the experiments are, our small test of them, delve into some things that made us a bit nervous but we conclude are not fatal flaws for intended purpose.

In the text, followed the Reviewer's suggestions (below) and revised these lines in the final paragraph of the introduction away from framing these as "questions" and rephrased the questions as statements. While we did not break the questions out of the paragraph and into bullets, we did number them to try and make them clearer to readers.

*The other three questions, which I call B1–B3, are written as follow-ups to be addressed by contributors and users of MIP simulation data:*

*B1 – Something about the robustness of the strength and width of the ITCZ.*

*B2 – Why do models disagree on "the relationship of climate sensitivity to base-state ITCZ width, even in sign."?*

*B3 – What is the mechanism "…of action for the consistent changes in ITCZ width that are generated by the addition of heat via surface fluxes?"*

*I think the paragraphs about these questions should be revised to make the questions clear—and the sentences themselves should be stated plainly as sentences. Maybe even express them in a bulleted list.*

*Also, maybe bullet the first two questions, and discuss them in a way that the reader can't mistake them for the three follow-on questions.*

Lines 315-330 are the final part of the last section of the paper; it is common to use this space to talk about what next steps follow from the research described in a manuscript. Here, we were trying to talk in some specifics that we think follow from the initial findings, that were not obvious to us when we set out on this investigation, mostly because the pilot MIP provides the beginnings of something concrete to work with. That's what we were trying to articulate here.

We've revised this to call these "avenues for further investigation" rather than questions, and revised these paragraphs overall to try and make them clearer.

Bulleted lists bring a lot of emphasis, visually, to their contents. We think they are best used sparingly in scientific papers, and that these were not the points we wanted to emphasize, so we leave them as in paragraph form. But we did revise the words that cue the three avenues we try and articulate.

*Figure 7: The "1e10" at the top of the colorbar would be clearer as "×10^10" (only formatted, without the ^).*

We will make this change.

*Citation: https://doi.org/10.5194/gmd-2024-17-RC1*

**Additional References**

Donohoe, A., D. M. W. Frierson, and D. S. Battisti, 2014: The effect of ocean mixed layer depth on climate in slab ocean aquaplanet experiments. *Climate Dynamics*, **43**, 1041–1055, https://doi.org/10.1007/s00382-013-1843-4.

---

## Author Comment (AC4)

**Response to community discussant comments from Wenyu Zhou**

*[Review comments in gray italic style*

Our responses in Normal text style]

*Nice work on intermodel comparison of the ITCZ influences. I enjoy reading the manuscript and have the following three comments.*

Thank you for the compliment and for your constructive comments.

*1. The investigation focuses on the annual mean. How about the effect on the seasonal timescales? Zhou et al. showed that future ITCZ contraction can induce a early-summer Hadley contraction, amid the annual-mean Hadley expansion.*

The experimental design and simulations don't include an annual cycle. So what we're analyzing should be thought of as more like perpetual single season's ITCZ (an equinoctal season).

Our response to your third comment is also relevant here.

We are adding citations to Zhou et al. (2019) and Zhou et al. (2020) in the revised manuscript.

*2. It may be clarified that the ITCZ width here is the annual-mean ITCZ width, which in essence reflects the extent of the seasonal migration of the ITCZ and is very different from the physical width of the ITCZ band when we view it at short (hourly, daily, monthly, and seasonal) timescales.*

An original motivation and goal was to look at the distribution of precipitation intensity in the ITCZ associated with width. We had forgotten to state this particular motivation in the original draft, and will add it in the revision (see also the response to Reviewer 3). But it is why we included daily precipitation, 500 hPa vertical velocity, near-surface specific humidity, which can be useful for analyzing precipitation on these timescales. We don't request shorter than daily timescale data because we wanted to keep storage requirements down and hopefully increase potential models contributing to the full MIP.

*3. The (annual-mean) ITCZ width is modified by perturbing the equatorial q-flux which changes the equatorial SST. I am thus surprised hat the paper did not mention the effect of SST pattern on the ITCZ width in the introduction (third paragraph around Line 25). As shown in Zhou et al., 2019, enhanced equatorial warming reduces the extent of the seasonal migration of ITCZ and narrows the annual-mean ITCZ width. Furthermore, the intermodel spread in the ITCZ change is largely explained by their different SST warming pattern. The SST warming pattern is a result of complex atmosphere-ocean coupled dynamics and cannot be attributed to the factors that the paper has listed.*

We don't address the seasonal cycle here (see our response to Reviewer 3 for more discussion of this decision), so what this MIP would focus on is somewhat orthogonal to these findings. Also, the focus of this metric clearly projects onto annual-mean ITCZ width, but seems to be mostly focused ITCZ location on seasonal timescales.

Part of our motivation for the persistent equinox configuration with peak meridional temperatures centered at the equator, rather than off the equator (like the QOBS profile) was to isolate the variations in width from potential interactions with location.

Since our simulations use a slab ocean and have fully interactive SSTs, they would capture any thermally-coupled interactions between the ITCZ and SST, in contrast to prescribed SST simulations, which have only unidirectional effects of SSTs on the atmosphere.

We don't claim or expect this experimental design to answer all questions. But what we would hope that further work applying this experimental design and carrying out more analysis focused on specific aspects of them is deeper understanding of some of the components that constitute the complex coupled dynamics of comprehensive simulations, such as those that Zhou et al., (2019) have documented. This seems to us like it may become a good example of where making use of a hierarchy of model (configurations) could deepen our understanding of our climate.

We've added a citation to Zhou et al., (2019).

**Additional References**

Zhou, W., S.-P. Xie, and D. Yang, 2019: Enhanced equatorial warming causes deep-tropical contraction and subtropical monsoon shift. *Nature Climate Change*, https://doi.org/10.1038/s41558-019-0603-9.

——, L. R. Leung, J. Lu, D. Yang, and F. Song, 2020: Contrasting Recent and Future ITCZ Changes From Distinct Tropical Warming Patterns. *Geophysical Research Letters*, **47**, e2020GL089846, https://doi.org/10.1029/2020GL089846.